# Movement Representation Strategies as a Tool for Educational Innovation in Physiotherapy Students: A Randomized Single-Blind Controlled-Pilot Trial

**DOI:** 10.3390/ijerph20054473

**Published:** 2023-03-02

**Authors:** Ferran Cuenca-Martínez, Luis Suso-Martí, Borja Peréz-Domínguez, Joaquín Calatayud, Rubén López-Bueno, Pedro Gargallo, María Blanco-Díaz, José Casaña

**Affiliations:** 1Department of Physiotherapy, University of Valencia, 46010 Valencia, Spain; 2Exercise Intervention for Health Research Group (EXINH-RG), Department of Physiotherapy, University of Valencia, 46010 Valencia, Spain; 3Department of Physical Medicine and Nursing, University of Zaragoza, 50009 Zaragoza, Spain; 4Department of Physiotherapy, Faculty of Medicine and Health Science, Catholic University of Valencia, 46001 Valencia, Spain; 5Surgery and Medical Surgical Specialties Department, Faculty of Medicine and Health Sciences, University of Oviedo, 33003 Oviedo, Spain

**Keywords:** educational innovation, physiotherapy, manual therapy, motor imagery, action observation

## Abstract

Physiotherapy has a strictly theoretical body of knowledge, but for the most part, the physiotherapist’s learning is practical. The practical part is fundamental to acquire clinical skills that the physiotherapist will later use in professional practice. The main aim of this study was to assess the effectiveness of movement representation strategies (MRS) in the improvement of manual skills of physiotherapy students as an educational innovation strategy. We randomly assigned 30 participants to an action observation practice (AOP), motor imagery practice (MIP), or sham observation (SO) group. A high velocity, low amplitude lumbar manipulation technique that is widely used in clinical physiotherapy practice was taught in one session. The primary outcomes were required time and test score. The secondary outcomes were perceived mental fatigue and perceived difficulty for learning. The outcomes were assessed preintervention and immediately after the intervention (postintervention). The main results showed that both AOP and MIP improved the total time required and the test score, as well as entailed less perceived difficulty for learning. However, both strategies showed a higher level of mental fatigue after the intervention, which was higher in the MIP group. Based on the results obtained, it seems that the application of MRS promotes greater learning of manual motor tasks in physiotherapy students and could be used as educational innovation strategies.

## 1. Introduction

Recently La Touche and Paris-Alemany [1] carried out a deep reflection and meta-cognitive work on the impact and possible benefit of implementing movement representation strategies (MRS) applied to the motor learning of clinical techniques in physiotherapy. Physiotherapy, like other health professions, has a strictly theoretical body of knowledge, but for the most part, the theoretical learning needs to be translated into practical skills. The practical part is fundamental to acquire clinical skills that the physiotherapist will later use in their professional practice [2]. Within procedural learning in physiotherapy, there is a wide range of manual techniques, such as mobilizations [3], manipulations [4], tractions [5], mobilizations with movement [6], etc., as well as different means to carry out a correct and complete implementation of a therapeutic exercise. So far, as La Touche and Paris-Alemany [1] comment, some methods have been developed for educational innovation regarding the teaching of manual techniques, such as experiential learning or simulation [7,8]. Even so, in the same line of La Touche and Paris-Alemany [1], we believe that the implementation of MRS could help to improve aspects related to motor learning that can be transferred to a greater learning of manual techniques in physiotherapy students.

Two of the most studied and used MRS to date are motor imagery practice (MIP) and action observation practice (AOP). MIP is defined as a dynamic mental process that involves the representation of an action, in an internal way, without its real motor action [9]. In addition, AOP evokes an internal, real-time simulation of what the observer is visualizing [10]. It has been shown that MIP and AOP may activate neurocognitive mechanisms underlying the planning, adjusting, and execution of voluntary movements in a similar way as when this movement is actually performed [11]. MRS have been studied in their application to motor learning processes. Previous research found that both MIP and OAP have shown a greater hand-learning process than a placebo [12,13]. Several hypotheses have been proposed [14], but it seems that processes of MRS share certain cerebral representations, along with processes of preparation for and real motor execution, suggesting that every action we execute is planned, initiated, and controlled through an imagery-like process [15,16]. Neuroimaging studies have revealed that, during MRS, there is a neurophysiological activation of the brain areas involved in the planning and execution of voluntary movement (primary motor cortex, supplementary motor area, cerebellum, premotor area, the inferior and superior parietal lobule, and the basal ganglia) similar to what happens when the movement is actually performed [17,18]. In addition, it had been suggested that OAP is more efficient, in terms of cognitive load, because the information provided through a video facilitates the job of the working memory, and, also, it does not need conscious strategies, compared to MIP [19].

Some work has been performed on SRMs in medical sciences or surgery [20]. So far, no research work has been applied to improve specific learning of clinical techniques in the field of physiotherapy. Recently Collet et al. [21] showed that medical students benefited from MIP training in clinical motor skills. The study conducted by Collet et al. [21] is very interesting and opens up a range of possibilities for the study of motor learning in health science students using MRS. We hypothesize that MRS can be a tool for learning clinical techniques in physical therapy students. In addition, based on previous studies [22], it is possible that the AOP could be an advantage in the initiation of the learning of these techniques. Therefore, than the main aim of this study was to assess the effectiveness of MRS in the improvement of the manual skills of physiotherapy students as a process of educational innovation. The secondary objective was to analyze the mental fatigue provoked by the MRS, as well as the difficulty perceived for learning.

## 2. Materials and Methods

### 2.1. Study Design

The present study was a randomized, single-blind placebo pilot trial, planned and conducted in accordance with Consolidated Standards of Reporting Trials requirements [23]. All procedures were approved by the Bioethics and Clinical Research Committee of UCH-CEU University (CEI20/004). All participants granted their written informed consent prior to inclusion and were provided an explanation of the study procedures, which were planned under the ethical standards of the Helsinki Declaration.

### 2.2. Participants

A total of 30 asymptomatic students were randomly assigned to the sham observation (placebo) group (SO, n = 10), MIP group (n = 10), or AOP intervention group (n = 10). The individuals were recruited from the UCH-CEU University in January 2022. The inclusion criteria were the following: (a) age >18 years-old, (b) be a student of the subject “Neuro-Orthopaedic Manual Physiotherapy”, (c) absence of pain at the time of the study, (d) understanding of Spanish language. The reason for not including participants with pain at the time of the study is because it has been shown that the presence of pain alters the ability to imagine, especially when the pain is persistent [24].

### 2.3. Randomization

Randomization was performed using a computer-generated random sequence table with a balanced 3-block design (GraphPad Software, Inc., San Diego, CA, USA). An independent researcher generated the randomization list, and a member of the research team who was not involved in the assessment or intervention of the participants was in charge of the randomization and maintained the list. Those included were randomly assigned to one of the three groups using the random-sequence list, ensuring concealed allocation.

### 2.4. Blinding

The assessments and interventions were performed by different physiotherapists (assistant professors). The evaluator was blinded to the participant’s allocation. All the intervention procedures were performed by the same physiotherapist, who had more than five years of experience in the field and was blinded to the purpose of the study. Participants were blinded to their group allocation.

### 2.5. Interventions

#### 2.5.1. First Phase: Motor Task

All the participants were verbally and visually instructed to learn how to perform a rotational lumbar manipulation technique. This high speed and low amplitude movement technique is an analgesic technique that is widely used in physiotherapy to improve low back pain [25,26,27]. This technique is learned through a motor sequence, as stated in the article conducted by Flynn et al. [4]. First, the patient is placed on the stretcher at an appropriate height depending on the characteristics of the therapist. Then the therapist is placed in a suitable position to be able to perform the technique. The technique is relatively simple: the patient is placed in lateral decubitus at the end of the table and with the legs bent. The therapist places the lower leg in pretension (slight traction). Subsequently, he/she performs an anterior–posterior mobilization to leave the position of the lower legs neutral. At the upper level, the patient takes hold of the therapist’s arm, and the therapist performs an anterior–posterior mobilization of the trunk in search of a neutral position. Once there, the therapist pulls the patient forward and applies two force vectors, one on each arm, causing a rotation of the lumbar spine due to the patient’s position. Once in this position, the therapist consumes the joint slack and, at the end of the range of motion, performs a low amplitude, high velocity impulse. This was the motor sequence performed and taught to the students (Figure 1).

#### 2.5.2. Second Phase: MRI

##### Motor Imagery

The MIP group performed a motor imagination task of a kinesthetic type and in first-person perspective. The participants had to imagine themselves performing the motor task they had previously performed in the classroom (first phase) during 10 series of 60 s, with a rest of 1 min every two series. The total duration of the intervention was 19 min. Other research studies with movement representation techniques had similar durations because continuous imagination can lead to mental fatigue [28,29]. They were told that no real movement should be made during the intervention. The participant’s imagined motor task was accompanied by the following phrases: “*keep imagining*”, “*focus your attention on the thigh*”, “*remember to imagine the task in the first person*”, and “*try to feel the movement*”. In addition, specific instructions were given according to the task to be imagined (Figure 2).

##### Action Observation

The AOP group performed an observation of the motor task. The participants had to observe the motor task they had previously performed in the classroom (first phase) during 10 series of 60 s, with a rest of 1 min every two series. The total duration of the intervention was 19 min. The participants were instructed to pay attention to all the steps performed in the technique in order to learn them and perform them again later. The observation was performed through a computer screen (Figure 2).

##### Sham/Placebo Observation

Participants in the SO group underwent a sham AOP protocol. The participants watched the material during the same intervention time as both previous groups. This documentary video was composed of video clips of nature landscapes, without any human agent or motor gesture. This type of sham AOP protocol has been used in previous studies [30,31] (Figure 2).

### 2.6. Procedures

After giving their consent to partake in the study and prior to the intervention, all participants were given a set of questionnaires. These included a sociodemographic assessment and an evaluation of their physical activity, mental chronometry (MC), and ability to imagine movements. The assessments were designed to have all participants start with the same mental state. The questionnaires were the Spanish-validated version of the International Questionnaire of Physical Activity (IPAQ) [32] and the Spanish-validated version of the Revised Movement Imagery Questionnaire (MIQ-R) [33]. The study was carried out in two sessions during the same day. In the first phase, all participants, at the same time, in a classroom, were instructed by an expert physiotherapist in the same way (verbal explanation and visual demonstration) and were allowed 20 min to practice the motor task in pairs with supervision. Once practiced, the evaluation of the main and secondary variables (preintervention assessment) was carried out. Once they were evaluated, in the second phase, the participants were divided into the three groups previously mentioned (MIP, AOP, or SO group) and the training of movement representation strategies was applied. At the end of these, the postintervention evaluation was carried out (postintervention assessment).

### 2.7. Outcomes

#### 2.7.1. Primary Outcomes

##### Required Time

The time required to perform the manipulation technique was recorded in seconds using a stopwatch.

##### Test Score

An evaluation template/sheet was used to address a score on several dimensions (patient position, therapist position, sequence, accuracy, and correct application of parameters), as well as a final score. This was the evaluation sheet used to assess the acquisition of manual skills of physiotherapy students in this subject. The scale presented a range of 0 to 10 points, where 0 points implies very poor performance and 10 points implies excellent performance.

#### 2.7.2. Secondary Outcomes

##### Perceived Mental Fatigue

We employed the Visual Analogue Scale of fatigue (VAS-f) to quantify the participants’ perceived fatigue after performing the training session. The VAS-f uses a line of 100 mm, with 0 representing minimum fatigue (no fatigue) and 100 representing maximum fatigue. The VAS-f scale is useful, sensitive, and easy to apply [34].

##### Perceived Difficulty

To assess the students’ perceived difficulty in learning the technique, a difficulty question (How difficult was the technique for you?) was asked. Students were asked to answer on a VAS scale of 100 mm, with 0 representing the minimum difficulty (no difficulty) and 100 representing the maximum difficulty.

#### 2.7.3. Baseline Outcomes

##### Mental Imagery Ability

To assess motor imagery ability, we employed the MIQ-R, which consists of four movements repeated in two domains (visual and kinesthetic). Depending on the perceived difficulty, participants score the movements from 1 to 7, with one representing the maximum difficulty in creating mental motor imagery and seven representing the least difficulty [33].

##### Mental Chronometry

MC is a reliable used tool for recording objective measurements of the ability to create mental motor images [35]. For the MC assessment, we used a stopwatch to record the time spent by each participant on executing and imagining the mental tasks included in the MIQ-R. The evaluator issued a command to start imagining the task, and the participant performed a verbal sign once the task had been completed. The time between the two interval commands was recorded, as was the time dedicated by each participant to the real-time execution of the task. The MC values are expressed as the time congruence between the two tasks (difference between imagination duration and execution duration) [35].

##### Physical Activity Level

We employed the self-reported IPAQ to assess the participants’ physical activity level. The total estimated resting expenditure energy (W/m^2^), or MET, was extracted [32].

### 2.8. Statistical Analysis

The statistical analysis was performed using SPSS software version 25.0 (SPSS Inc., Chicago, IL, USA). The normality of the variables was evaluated using the Shapiro–Wilk test. Descriptive statistics were used to summarize the data for the continuous variables and are presented as mean ± standard deviation and 95% confidence interval. A two-way repeated measures analysis of variance (ANOVA) was conducted to study the effect of the between-subject factor ‘intervention group’ with 3 categories (MIP, AOP, and SO groups) and the within-subject called ‘time’ with also 2 categories (pre- and postintervention) on the dependent variables. Partial eta squared (ƞ_p_^2^) was calculated as a measure of effect size for each main effect and interaction in the ANOVAs, with 0.01–0.059 representing a small effect, 0.06–0.139 a medium effect, and >0.14 a large effect [36]. A post hoc analysis with Bonferroni correction was performed in the case of significant ANOVA findings for multiple comparisons between variables. Effect sizes (*d*) were calculated according to Cohen’s method, in which the magnitude of the effect was classified as small (0.20–0.49), moderate (0.50–0.79), or large (0.8) [37]. The α level was set at 0.05 for all tests.

## 3. Results

A total of 30 participants were included and were randomly allocated into three groups of 10 participants per group. There were no adverse events reported in either group. All the variables presented a normal distribution. No statistically significant differences were found between groups for any of the demographic data or self-report variables that were present at baseline between the groups (Table 1).

### 3.1. Main Results

#### 3.1.1. Time Required

The ANOVA revealed significant changes in the time required measurement during group*time interaction (F = 4.37, *p* = 0.023, ƞ2 = 0.245) and time (F = 13.96, *p* < 0.01, ƞ2 = 0.241). The post hoc analysis revealed significant between-group and intragroup differences (Table 2). Statistically significant differences were observed between the preintervention assessment and the postintervention assessment in the AOP and MIP groups, with a large effect size (*p* < 0.001, d = 2.27 and d = 1.01, respectively) (Figure 3). In addition, statistically significant between-group differences were observed between the AOP and MIP groups and the SO group in the postintervention assessment, with a large effect size in favor of the AOP and MIP (less time required) (*p* < 0.001, d = 1.77 and d = 1.54, respectively).

#### 3.1.2. Test Score

The ANOVA revealed significant changes in the time required measurement during group*time (F = 3.37, *p* = 0.013, ƞ2 = 0.225) and time (F = 15.06, *p* < 0.01, ƞ2 = 0.234). The post hoc analysis revealed significant between-group and intragroup differences (Table 2). Statistically significant differences were observed between the preintervention assessment and the postintervention assessment in all groups, with a large effect size (*p* < 0.001, AOP group d = 3.94, IM group d = 2.77, SO group d = 1.04) (Figure 4). In addition, statistically significant between-group differences were observed between the AOP and MIP groups and the SO group in the postintervention assessment, with a large effect size in favor of the AOP and MIP (better test score) (*p* < 0.001, d = 2.77 and d = 2.07, respectively).

#### 3.1.3. Perceived Difficulty

The ANOVA revealed significant changes in the time required measurement during group*time (F = 40.03, *p* < 0.001, ƞ^2^ = 0.748) and time (F = 180.96, *p* < 0.01, ƞ^2^ = 0.871). The post hoc analysis revealed significant between-group and intragroup differences (Table 2). Statistically significant differences were observed between the preintervention assessment and the postintervention assessment in the AOP and MIP groups, with a large effect size (*p* < 0.001, d = 3.55 and d = 2.77, respectively) (Figure 5). In addition, statistically significant between-group differences were observed between the AOP and MIP groups and the SO group in the postintervention assessment, with a large effect size in favor of the AOP and MIP groups (less difficulty) (*p* < 0.001, d = 2.98 and d = 1.82 respectively). Finally, statistically significant between-group differences were observed between the AOP and IM groups in favor of the AOP (less difficulty) (*p* < 0.001, d = 1.70).

#### 3.1.4. Mental Fatigue

The ANOVA revealed significant changes in the time required measurement during group*time (F = 34.22, *p* < 0.001, ƞ^2^ = 0.556) and time (F = 80.66, *p* < 0.01, ƞ^2^ = 0.705). The post hoc analysis revealed significant between-group and intragroup differences (Table 2). Statistically significant differences were observed between the preintervention assessment and the postintervention assessment in the AOP and MIP groups, with a large effect size (*p* < 0.001, d = 2.25 and, d = 3.44, respectively) (Figure 6). In addition, statistically significant between-group differences were observed between the AOP and MIP groups and the SO group, with a large effect size in favor of the SO group (less mental fatigue) (*p* < 0.001, d = 1.22 and d = 2.88, respectively). Finally, statistically significant between-group differences were observed between the AOP and IM groups in favor of the AOP (less mental fatigue) (*p* < 0.001, d = 2.01).

## 4. Discussion

The main aim of this study was to assess the effectiveness of movement representation strategies in the improvement of manual skills of physiotherapy students as a process of educational innovation. The secondary objective was to analyze the mental fatigue provoked by the MRS, as well as the perceived difficulty. The main results showed that both AOP and MIP improved the total time required and the test score, as well as entailed a better perceived difficulty for learning. However, both strategies showed a higher level of mental fatigue after the intervention, which was higher in the MIP group.

The results obtained in our study are in line with other research studies. For example, Cuenca-Martinez et al. [22] found that both OAP and MIP in isolation elicited a motor learning process of a motor sequence of manual tasks. Moreover, AOP showed the most robust changes, which were maintained for at least 16 weeks postintervention. These results were also found in people who performed a sequence of lumbo–pelvic control motor tasks [38]. The results of the study by Cuenca-Martínez et al. [38] showed that, by adding AOP to a motor control exercise program, motor control exercises were learned faster compared to a control group. Regarding the direct comparison between MIP and AOP, in the study conducted by Gatti et al. [19], the participants had to learn a complex motor task that involved moving the right hand and foot in the same angular direction, while simultaneously moving the left hand and foot in an opposite angular direction. The authors also found that AOP was slightly more effective than MIP. In addition, González-Rosa et al. [13] also found that AOP was more effective than MIP in promoting the early learning of a new complex coordination task. At the educational learning level, the research work conducted by Collet et al. [21] showed that the application of MRS, in combination with physical practice, resulted in faster learning (in fewer sessions) than just physical practice alone. These results are in line with the results found in this research, showing that the application of MRS obtained better results than the real isolated practice, and, furthermore, that the AOP obtained better results in comparison with MIP.

We should consider that our results reflect a motor learning process through MRS. In addition, there is abundant scientific evidence that tells us that MRS promotes human motor learning. We cannot forget that there are already systematic reviews with meta-analyses that show that applying MRS to an intervention promotes the improvement of motor learning and prevents motor deconditioning in different clinical variables of interest in immobilized or non-immobilized patients, whether or not they have undergone surgery [20,39]. We are therefore in front of some tools of great potential that can be used for learning motor tasks in the field of health sciences. It seems that the performance of a motor task in a physical manner shares a common neurophysiological substrate with the application of MRS. These brain areas are diverse, but perhaps those responsible for planning, adjustment, and preparation for voluntary movement, among others, should be stressed. Some variables may influence the effectiveness of MRS. For example, previous physical condition, imaginative ability, mental fatigue, or self-efficacy expectations may modulate these effects [29]. AOP seems to be slightly more effective and efficient, because only the input image has to be processed, and not created and maintained, as is the case during MIP. This may be the reason that MIP causes higher levels of fatigue, which could be related to a lower effectiveness in learning a complex gesture such as the one used in this study. However, MIP can modulate the context and environment on the demand of the intervener, and this is a clear advantage [29]. It should also be noted that all three groups obtained improvements in the test score at postintervention, so we cannot rule out the existence of a consolidation component external to the MRS (perhaps because they had already performed the technique once before in preintervention), since this also occurred even in the SO (although to a lesser extent). Finally, both MRS should be considered for the acquisition of new motor gestures, in combination, combined with physical practice and in isolation, depending on the context [29].

### Study Limitations

This study has some limitations that should be taken into consideration. For example, the intervention was only one day. It would have been very interesting to evaluate not only the rapid motor learning phase but also some consolidation and automatization phases. Future studies should aim to lengthen the days of intervention. It is also important to stress that the evaluation was immediately after the intervention. It would have been very interesting to make more measurements over time in order to observe whether or not learning is maintained over time. This should be addressed in future studies. Finally, this is a pilot study, and the results should be viewed with caution. This research work should serve to show preliminary data; however, future studies should contain a larger sample (based on a size calculation for an alpha error of 5% and a beta error of between 5 and 20%) to see slightly more robust results.

## 5. Conclusions

Based on the results obtained, it seems that the application of movement representation strategies promotes greater learning of manual motor tasks in physiotherapy students compared to not applying these techniques. Movement representation strategies are low cost and can be used as educational innovation strategies for the improvement of the performance of manual clinical techniques in physiotherapy (university environment). It appears that action observation training is slightly more effective than motor imagery practice, and both are superior to a sham intervention. Future research should be aimed at trying to evaluate when it is more appropriate to choose one technique or the other depending on the context.

## Figures and Tables

**Figure 1 ijerph-20-04473-f001:**
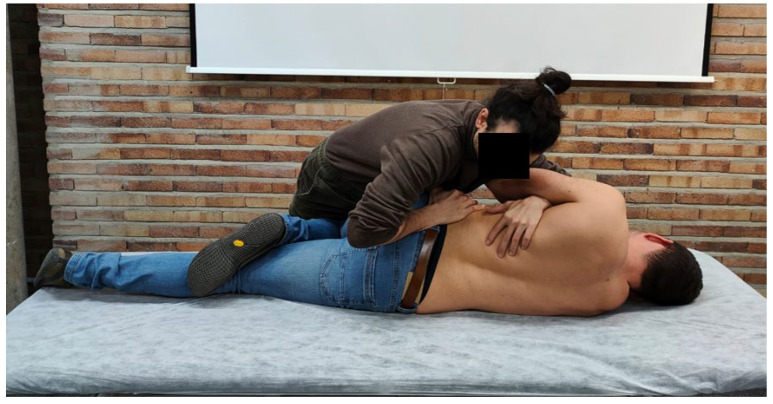
Motor task of rotational lumbar manipulation technique.

**Figure 2 ijerph-20-04473-f002:**
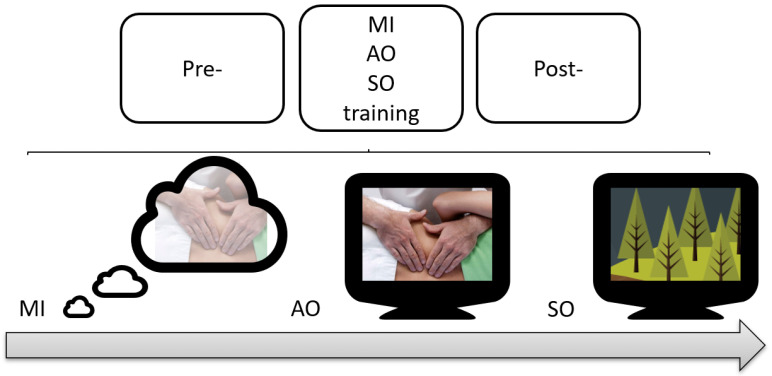
An illustration of the second phase interventions.

**Figure 3 ijerph-20-04473-f003:**
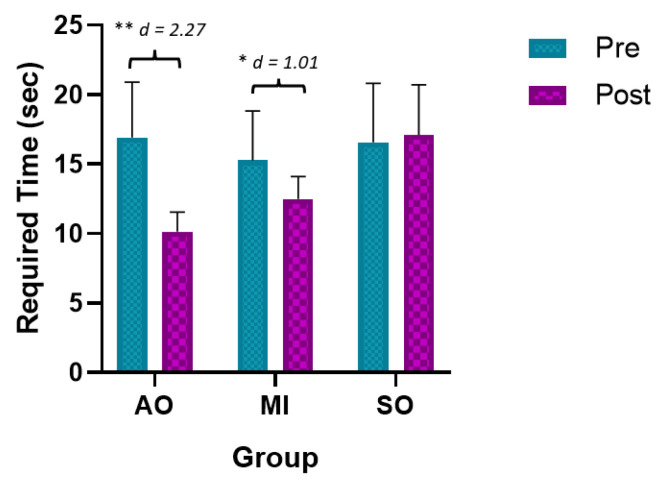
Main results for required time. * *p* < 0.05; ** *p* < 0.01; MIP: motor imagery practice; AOP: action observation practice; SO: Sham observation.

**Figure 4 ijerph-20-04473-f004:**
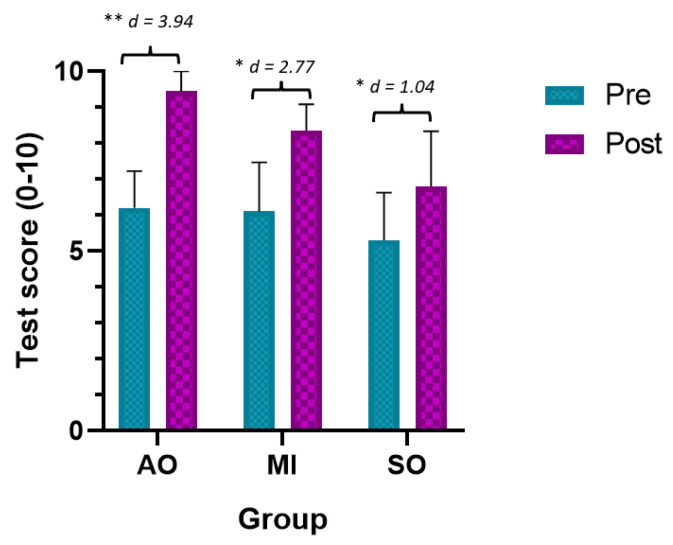
Main results for test score. * *p* < 0.05; ** *p* < 0.01; MIP: motor imagery practice; AOP: action observation practice; SO: Sham observation.

**Figure 5 ijerph-20-04473-f005:**
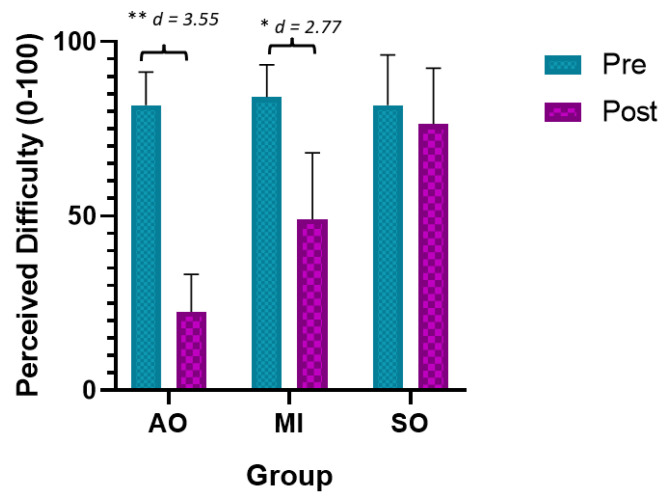
Main results for perceived difficulty. * *p* < 0.05; ** *p* < 0.01; MIP: motor imagery practice; AOP: action observation practice; SO: Sham observation.

**Figure 6 ijerph-20-04473-f006:**
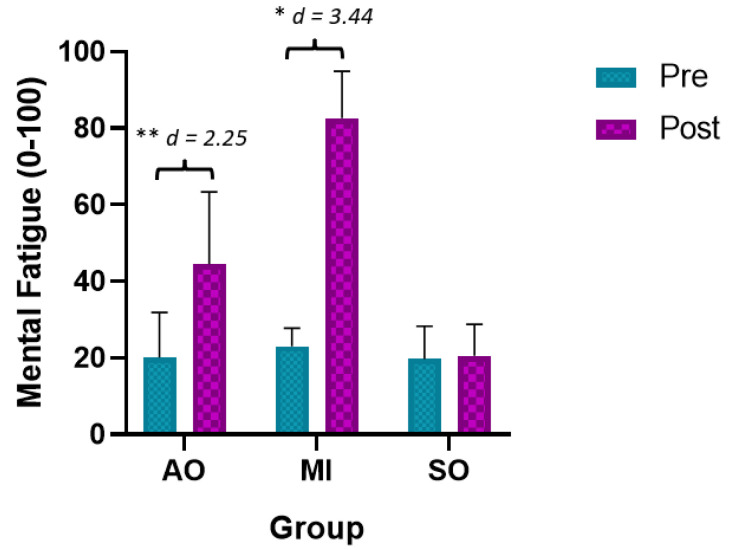
Main results for mental fatigue. * *p* < 0.05; ** *p* < 0.01; MIP: motor imagery practice; AOP: action observation practice; SO: Sham observation.

**Table 1 ijerph-20-04473-t001:** Descriptive statistics of sociodemographic and baseline data.

Measures	AOP Group (n = 10)	MIP Group (n = 10)	SO Group (n = 10)	*p* Value
Age	21.5 ± 0.55	21.6 ± 2.53	21.70 ± 1.39	0.920
Height (cm)	172.9 ± 0.80	167.10 ± 0.70	179 ± 0.40	0.698
Weight (Kg)	66.3 ± 4.97	68.70 ± 5.3	67.5 ± 4.26	0.552
Gender				0.555
Male	4 (40)	3 (30)	4 (40)	
Female	6 (60)	7 (70)	6 (60)	
IPAQ	1660.6 ± 283.51	1813.85 ± 500.3	1725.2 ± 559.17	0.988
MIQ-R	46.3 ± 4.57	47.3 ± 3.76	48.2 ± 5.52	0.760
MC	3.55 ± 2.96	2.39 ± 2.7	4.71 ± 3.22	0.809

Values are presented as mean ± standard deviation or number (%); MIP: motor imagery practice; AOP: action observation practice; SO: Sham observation.

**Table 2 ijerph-20-04473-t002:** Main results obtained.

Measure	Group			
Required Time		Pre	Post	Mean Difference (95% CI); Effect Size (*d*)
	AOP	16.93 ± 3.98	10.14 ± 1.42	−6.78 ** (−9.95 to −3.62); *d* = 2.27
	IMP	15.28 ± 3.55	12.49 ± 1.63	−2.79 * (−5.95 to −0.37); *d* = 1.01
	SO	16.57 ± 4.26	17.11 ± 3.61	0.4 (−3.56 to 2.76); *d* = -
Mean Difference (95% CI); Effect Size (*d*)	(a) AOP vs. IM (b) AOP vs. SO (c) IMP vs. SO		(a) −0.35 (−3.2 to 2.51); *d* = . (b) −3.77 ** (−6.63 to −0.91); *d* = 1.77 (c) −3.42 * (−6.29 to −0.56); *d* = 1.54	
Test Score		Pre	Post	Mean Difference (95% CI); Effect Size (*d*)
	AOP	6.2 ± 1.03	9.45 ± 0.55	−3.25 ** (−4.25 to −2.25); *d* = 3.94
	IMP	6.1 ± 1.37	8.35 ± 0.74	−2.25 * (−3.25 to −1.24); *d* = 2.77
	SO	5.3 ± 1.33	6.8 ± 1.54	−1.5 * (−2.5 to −0.49); *d* = 1.04
Mean Difference (95% CI); Effect Size (*d*)	(a) AOP vs. IM (b) AOP vs. SO (c) IMP vs. SO		(a) −0.62 (−0.38 to 1.58); *d* = - (b) −1.78 ** (0.79 to 2.75); *d* = 2.77 (c) −1.17 * (0.19 to 2.15); *d* = 2.07	
Perceived Difficulty		Pre	Post	Mean Difference (95% CI); Effect Size (*d*)
	AOP	81.8 ± 9.53	22.5 ± 10.8	−59.3 ** (−50.54 to −68.05); *d* = 3.55
	IMP	84.1 ± 9.31	49 ± 19.14	−35.1 ** (−43.85 to −25.35); *d* = 2.77
	SO	81.8 ± 14.43	76.4 ± 15.94	0.4 (−3.56 to 2.76); *d* = -
Mean Difference (95% CI); Effect Size (*d*)	(a) AOP vs. IM (b) AOP vs. SO (c) IMP vs. SO		(a) −0.26.5 ** (−44.41 to −8.58); *d* = 1.70 (b) −53.9 ** (−71.81 to −35.98); *d* = 2.98 (c) −27.4 * (−45.31 to −9.48); *d* = 1.82	
Mental Fatigue		Pre	Post	Mean Difference (95% CI); Effect Size (*d*)
	AOP	20.1 ± 11.74	44.5 ± 18.9	24.4 ** (−36.04 to −12.75); *d* = 2.25
	IMP	23 ± 4.83	82.6 ± 12.30	−59.6 ** (71.25 to −47.95); *d* = 3.44
	SO	19.7 ± 8.58	20.5 ± 8.31	4.5 (−16.14 to 7.15); *d* = -
Mean Difference (95% CI); Effect Size (*d*)	(a) AOP vs. IM (b) AOP vs. SO (c) IMP vs. SO		(a) −0.38.1 ** (−59.95 to −22.25); *d* = 2.01 (b) 24 ** (8.14 to 39.85.98); *d* = 1.22 (c) 61.1** (46.25 to 77.95); *d* = 2.88	

* *p* < 0.05; ** *p* < 0.01; MIP: motor imagery practice; AOP: action observation practice; SO: Sham observation; CI: Confidence Interval.

## Data Availability

Not applicable.

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
