# Peer review of "Movement Representation Strategies as a Tool for Educational Innovation in Physiotherapy Students: A Randomized Single-Blind Controlled-Pilot Trial"

_ijerph, 2023, doi:10.3390/ijerph20054473_

Round 1

Reviewer 1 Report

Short Summary:

The study investigated whether action imagery practice or action observation practice that is added after a primary physical exercise improves motor learning of a lumbar manipulation technique that is widely used in physiotherapy. The results indicate that action imagery practice as well as action observation practice evolved in further improvements whereas as control group (watching landscape videos) showed lower improvements.

General Comments

The manuscript investigates an interesting research question. However, particularly the research question and theoretical deepness may be developed in more detail in the introduction and discussion. Below, I will provide direct and hopefully helpful suggestions to improve the manuscript.

To me it appeared a little awkward that more than half of the references come from the authors own research group, whereas other very important research groups in action imagery theory are missing. There is a reference citing David Wright and another one citing Jean Decety. However, it might also be fruitful to integrate the fundamental works from Marc Jeannerod, Aymeric Guillot, Cornelia Frank, Martina Rieger, and Shaun Boe. All of them have had an impressive impact on motor imagery theory. Thereby, I would like to mention a very recent special issue in Psychological Research on motor imagery theories, which might be helpful to give the manuscript more theoretical depths. There will be at least 4 theoretical issues coming out recently or soon (Bach et al., 2022, Frank et al., 2022, Krueger et al., 2022, Rieger et al., 2023).

The primary exercise should be explained in more detail. Were there any instructions (written, verbal, videos) given ahead? Did participants observe someone doing the action before trying it themselves? How did the participants know what to exercise? The primary exercise is the first time mentioned in the methods (line 115), but should already be considered in the introduction as this is an essential part of the study. Please explain why the action was practically performed in advance. Which is the theoretical reasoning for doing so?

From the introduction it remains unclear to me why there was no action execution practice group involved in the study (a group that actually performs the movement while the others imagine / observe). Maybe explain in more detail in the introduction what are the (expected) advantages of learning via imagination / observation (instead of as well as in addition to execution).

The last paragraph in the introduction lacks a clear explanation of the expectations. Please make sure to state 1. What is expected in which group in regards to each dependent variable? 2. Why is it expected?

Please provide an ad-hoc sample size estimation (line 88). n=10 seem quite low to me. If this study is just a pilot study (line 358), the second main study should be added to the manuscript!

Figure 4. The text states that there was also a sign. difference between pretest and posttest in the SO group. Why is there no such indication in the figure? This finding should be discussed in the discussion! Why did they improve? Consolidation (although distracted by videos)?

Figure 3 does not include variance information, which is essential for inferential statistics! I’d suggest do provide boxplots with jitter instead of bar graphs. This accounts for all statistical graphs.

I personally found the finding of mental fatigue (Fig. 6) quite interesting. This maybe discussed in more detail in the discussion.

Differences between AOP and AIP should be discussed in more detail. In the discussion (line 344-349) as well as in the introduction.

Minor Comments

-       I personally prefer the term action imagery practice and action observation practice as they fit nicely together. The authors may consider these terms as well. Otherwise, it remains unclear why motor imagery is not called ‘training’, but action observation training is called a ‘training’. (e.g. line 51)

-       line 27: what does ‘better’ perceived difficulty mean? Rated less difficult?

-       line 87: delete ‘recruitment of’ from the ‘participants’ header

-       line 93: delete the exclusion criteria. Instead place another inclusion criteria: d) Spanish [a and c were the same thing]

-       line 114: ‘everything’ The reader does not know what that means. Please explain the action in detail here. I believe that a picture or a sequence of pictures might be helpful that indicates the action

-       line 114: the participant à participants … themselves

-       line 125: Figures should be placed after the first mentioning. The action itself does not come clear from this. Is there a third arm in the upper right corner of the picture? Why? Maybe replace this figure by a clear illustration of the action? I am not a physiotherapist (as other readers will be) and would greatly benefit from an illustration that explains what is explained in lines 153-163.

-       Maybe combine interventions and procedure into one single header which might also include randomization and blinding.

-       line 116-118: 10 x 1min practice + 9 x 1min rest = 19 minutes. Where do the 15 minutes come from?

-       line 127ff: When explaining action observation practice focus more on the subject itself (AOP), rather than explaining action imagery practice (which has been explained first hand)

-       line 131: ‘performed’ : does this mean executed? Consider another term or rephrasing.

-       line 134-137: consider rewriting and shortening this part

-       line 165: delete: ‘without group distinction’

-       line 175f: Is it considered good to be fast? I guess that many therapeutical movements should be done slowly. Please explain in the text.

-       line 178ff: please provide the scales for the test score: 1 excellent to 10 very poor? This is not clear for the reader at this point.

-       Place the mental chronometry measure after the MIQ-R as it is based on that.

-       line 202-204: I did not really understand why inter-rater correlations and ICC were calculated here. Further, did participants give verbal signs for ‘time measurement’ during execution too? Please state this here. How was the ‘time congruence’ calculated? Is it the difference imagination duration – execution duration? Please make that clear in the text. Just out of interest: Was there any correlation between MIQ scores and ‘time congruence’?

-       delete lines 210-212: it appears to me that this is not based on the own data. Are you aware that there is a more actual version of the MIQ, which is the MIQ-3 https://www.mdpi.com/2077-0383/11/20/6076 and Williams, S.E.; Cumming, J.; Ntoumanis, N.; Nordin Bates, S.M.; Ramsey, R.; Hall, C. Further validation and development of the movement imagery questionnaire. J. Sport Exerc. Psychol. 201234, 621–646.

-       line 215: How was ‘MET extracted’? What kind of ratings did participants give here?

-       delete line 216-218, which are not based on the own data

-       line 228: delete ‘strength of association’

-       line 239: ‘primary variables’ should probably not be placed here. Later there are sign. differences shown in the results!

-       Figure 2 is not necessary. Consider dropping it.

-       Table 1: refer to it in the participants section and place it there

-       all results: Please provide degrees of freedom (see APA style)!

-       line 250: Please explain in detail in the text, not in table (Table 2). If there were sign. differences: Please state which group had higher values than the other, instead of just mentioning a difference. 

-       line 247-256: The group effect has not been described here. Always provide n.s. effects too (see APA style)!

-       line 324-329: Please state clearer what these results have to do with theoretical conclusions that come from your own research.

-       line 330f: This is not a review article. Please discuss the presented results in the discussion, not those of other studies.

Reviewer 2 Report

This is a worthwhile study in the field of Physiotherapy didactics that is relevant for teaching practice.

Two issues are required to be changed:

1) There are many typos in the manuscript, especially also in the abstract. This needs to be edited.

2) There is quite probounced self citation in the text. I strongly recommend to balance this more and not to exaggerate own contributations.
